# Neurobiological Correlates of Coping Strategies in PTSD: The Role of IGF-1, CASP-9, nNOS, and IL-10 Based on Brief-COPE Assessment

**DOI:** 10.3390/cimb47100868

**Published:** 2025-10-21

**Authors:** Barbara Paraniak-Gieszczyk, Ewa Alicja Ogłodek

**Affiliations:** Collegium Medicum, Jan Dlugosz University in Częstochowa, Waszyngtona 4/8 Street, 42-200 Częstochowa, Poland

**Keywords:** Brief-COPE, CASP-9, coping strategies, IGF-1, IL-10, nNOS, oxidative stress, post-traumatic stress disorder, resilience, trauma

## Abstract

Post-traumatic stress disorder (PTSD) is associated with long-term disturbances in stress regulation, neuroinflammation, and oxidative stress and reduced psychological coping capacity. The aim of the study was to assess the relationship between selected neurobiological biomarkers (Insulin-like Growth Factor 1—IGF-1; Caspase-9—CASP-9; Neuronal Nitric Oxide Synthase—nNOS; and Interleukin-10—IL-10) and coping styles evaluated using the Brief Coping Orientation to Problems Experienced (Brief-COPE) questionnaire in men with trauma experience. Particular emphasis was placed on analyzing the effect of PTSD chronicity (≤5 years vs. >5 years) on these relationships. The study included 92 adult men with a history of life-threatening situations. Participants were divided into three groups: PTSD within the past ≤5 years (*n* = 33), PTSD within the past >5 years (*n* = 31), and a No PTSD group (*n* = 28). Biomarkers were measured in blood serum. Coping strategies were assessed using the Brief-COPE questionnaire, which includes four subscales: task-oriented, emotion-oriented, avoidant, and general coping. Due to the lack of normal distribution, the Kruskal–Wallis test and Dunn’s post hoc test were used. Correlations between biomarkers and Brief-COPE subscales were calculated using Spearman’s Rank Correlation Coefficient (Rho). Significant differences between groups were found in all four biomarkers (*p* < 0.001). IGF-1 and IL-10 reached the highest values in the No PTSD group and the lowest in the PTSD ≤ 5 years group, indicating neuroprotective and anti-inflammatory deficits in PTSD. Conversely, CASP-9 and nNOS levels (markers of apoptosis and oxidative stress) were highest in PTSD ≤ 5 years, with partial normalization in the PTSD > 5 years group. In terms of coping strategies, the No PTSD group displayed a highly adaptive profile (task-oriented: 30/32; emotion-oriented: 43/48; and avoidant: 12/32). Individuals with PTSD ≤ 5 years presented a maladaptive pattern (task-oriented: 13/32; avoidant: 26/32; and emotion-oriented: 27/48), while in PTSD > 5 years, a further decline in emotion-oriented (21/48) and general coping (59/112) was observed, suggesting progressive depletion of psychological resources. The strongest correlations between biomarkers and coping strategies occurred in PTSD groups. Low IGF-1 levels in PTSD ≤ 5 years correlated negatively with emotion-oriented coping (Rho = −0.39) and general coping (Rho = −0.35). High CASP-9 levels were associated with reduced task-oriented coping in PTSD > 5 years (Rho = −0.29). Similar trends were observed for nNOS and IL-10, indicating a disturbance in neurobiological balance that favors persistence of PTSD symptoms. PTSD, both in its acute and chronic phases, is associated with an abnormal profile of neuroprotective, apoptotic, and inflammatory biomarkers, which correlates with impaired adaptive coping capacity. Although partial normalization of biological parameters is observed in chronic PTSD, deficits in emotion-oriented and task-oriented coping persist. The Brief-COPE questionnaire, combined with biomarker analysis, may serve as a useful clinical tool for assessing psychophysiological balance and designing early interventions. These results highlight the potential of IGF-1, CASP-9, nNOS, and IL-10 as biomarkers of stress adaptation and therapeutic targets in PTSD.

## 1. Introduction

Post-traumatic stress disorder (PTSD) is a serious mental disorder that develops as a result of severe or prolonged exposure to stress exceeding an individual’s adaptive capacity [1,2,3]. From a neurobiological perspective, it is associated with persistent changes in the functioning of brain structures responsible for regulating emotions, memory, and stress responses—particularly the amygdala, hippocampus, and prefrontal cortex [4,5,6,7]. This disorder is also characterized by dysregulation of the Hypothalamic–Pituitary–Adrenal (HPA) axis, excessive sympathetic activation, and intensified inflammatory responses in the central nervous system. Understanding the neurobiological mechanisms of PTSD is crucial for developing more effective therapeutic strategies and identifying individuals particularly vulnerable to developing the disorder as a result of prolonged stress [8,9,10]. According to the *Diagnostic*
*and Statistical Manual of Mental Disorders, Fifth Edition* (*DSM-5*), PTSD includes four main groups of symptoms: intrusions (e.g., intrusive memories and nightmares), avoidance (e.g., avoiding thoughts or conversations about the trauma), cognitive–emotional disturbances (e.g., negative self-beliefs and feelings of alienation), and hyperarousal (e.g., irritability, hypervigilance, and sleep difficulties) [11]. Despite advances in understanding the pathophysiology of PTSD, many biological mechanisms remain unclear, and the effectiveness of both pharmacological and psychotherapeutic treatment is still unsatisfactory for all patient groups [12,13,14]. Therefore, further research integrating biological and psychological mechanisms of PTSD remains essential, taking into account factors such as symptom duration and coping style. Modern neurobiology of PTSD indicates that post-traumatic stress affects the functioning of the entire nervous system—leading to HPA axis dysregulation, sympathetic overactivation, microglial reactivity, and altered neurotransmitter metabolism and neuroplasticity [15,16,17,18]. Chronic stress also promotes the production of pro-inflammatory cytokines, free radicals, and the intensification of apoptotic processes within neurons [19].

In this context, increasing importance is attributed to biomarkers reflecting various aspects of the biological response to stress. Significant biomarkers include: Insulin-like Growth Factor 1 (IGF-1), caspase-9 (CASP-9), neuronal Nitric Oxide Synthase (nNOS), and interleukin-10 (IL-10). Each of these markers represents a distinct biological mechanism: IGF-1 is responsible for neurotrophic activity and neuroregeneration [20]; CASP-9 for apoptosis and mitochondrial stress [21]; nNOS for neurotransmission regulation and nitrosative stress [22]; and IL-10 for controlling neuroinflammatory immune responses [23]. Insulin-like growth factor-1 plays a key role in neurogenesis, neuronal differentiation and survival, and protection against oxidative stress and glutamate excitotoxicity [24]. Reduced IGF-1 levels observed in patients with PTSD may indicate a weakened ability of the brain to regenerate and adapt. Studies have shown that IGF-1 deficiency correlates with increased depressive symptoms, cognitive impairments, and amygdala hyperactivity—suggesting IGF-1 involvement in regulating anxiety responses and the stress axis [25,26,27]. Caspase-9 is a key enzyme initiating the mitochondrial pathway of apoptosis. Its activation in response to cellular stress leads to the release of cytochrome c and activation of the caspase cascade, resulting in neuronal death [28,29,30]. In PTSD models, increased CASP-9 expression has been observed in limbic and cortical structures associated with emotional regulation, correlating with intensified neurodegeneration and cognitive dysfunctions [31,32]. There is also evidence that CASP-9 may influence synaptic plasticity and the generation of Reactive Oxygen Species (ROS), making it a potential target for therapeutic interventions [33,34]. Neuronal nitric oxide synthase participates in the regulation of synaptic functions, cerebral blood flow, and neurogenesis. However, its excessive activity under stress conditions leads to the production of Reactive Nitrogen Species (RNS), damaging lipids, proteins, and DNA [35,36]. In animal models of PTSD, increased nNOS expression has been observed in limbic structures, associated with anxiety and depressive symptoms [37,38]. Furthermore, nNOS may interact synergistically with pro-inflammatory cytokines and glucocorticoids, leading to microglial activation, blood–brain barrier damage, and neuronal apoptosis [39,40]. Interleukin-10 is the main anti-inflammatory mediator in the central nervous system [41,42,43]. Its role is to inhibit pro-inflammatory cytokine expression and limit microglial activation. Reduced IL-10 levels have been observed in some PTSD studies, suggesting weakened anti-inflammatory control, although results remain inconsistent. Such a state promotes the persistence of a neuroinflammatory environment and may contribute to hippocampal and prefrontal cortex dysfunction. The cytokine profile in PTSD often shows increased pro-inflammatory activity (e.g., IL-6 and TNF-α) accompanied by insufficient anti-inflammatory regulation, associated with greater psychopathological symptoms, sleep disturbances, and higher risk of somatic diseases [44,45,46]. Alongside biological aspects of PTSD, psychological coping strategies are also of great importance. According to Lazarus and Folkman’s model, coping is a cognitive and behavioral process through which an individual attempts to deal with situations perceived as stressful or exceeding their resources [47,48]. The Brief-COPE questionnaire allows assessment of three main coping styles: task-oriented (focused on problem-solving), emotion-oriented (expressing and experiencing emotions), and avoidant (denial, escape, and distancing). Research shows that a task-oriented style promotes better adaptation, whereas an avoidant style may exacerbate PTSD symptoms and contribute to its chronic course [49].

Despite growing interest in integrating neurobiological and psychological approaches, direct analyses of relationships between stress biomarkers and coping styles remain limited. Particularly little is known about how PTSD duration—for example, immediately after trauma versus many years later—affects these relationships. Understanding the links between neurobiological stress markers and coping strategies, with consideration of PTSD chronicity, may provide valuable insights into individual differences in the course of the disorder and help tailor therapeutic interventions more effectively to patients’ needs.

The aim of this study was to analyze the relationships between IGF-1, CASP-9, nNOS, and IL-10 levels and coping strategies assessed with Brief-COPE in men with a history of traumatic experiences, taking into account PTSD duration. It was assumed that the levels of these biomarkers not only reflect the neurobiological state of the individual but are also associated with coping style. The analysis also includes an assessment of whether PTSD chronicity is linked to lasting changes both biologically and psychologically.

## 2. Materials and Methods

### 2.1. Characteristics of the Participants

Ninety-two men aged 18 to 50, professionally exposed to extreme stress in the working conditions of mine rescue workers who had experienced life-threatening traumatic events, were enrolled in the study. Participants were divided into three groups depending on PTSD status: 33 individuals (35.9%) with diagnosed PTSD within 5 years of the traumatic event (≤5 years), 31 individuals (33.7%) with PTSD persisting for more than 5 years after trauma (>5 years), and 28 individuals (30.4%) comprising the No PTSD group—without current or past symptoms of PTSD. The No PTSD group included trauma-exposed individuals who did not develop PTSD.

Group assignment was carried out by a specialist in psychiatry and family medicine—a co-author of this article. The inclusion criteria for the study groups were: male sex, age 18–50 years, occupational stress exposure, traumatic experience, and PTSD diagnosis consistent with DSM-5 criteria, confirmed on the basis of a clinical interview and the Clinician-Administered PTSD Scale for DSM-5 (CAPS-5 scale). The No PTSD group consisted of healthy men without symptoms or diagnosis of PTSD, matched by age to the study groups.

Exclusion criteria were: presence of other psychiatric disorders (except PTSD in the study groups), somatic diseases, current medication use, nicotine or other psychoactive substance dependence (including drugs, narcotics, and alcohol), legal incapacitation, and employment in uniformed services (military and police). Thanks to these criteria, a relatively homogeneous group of men at high risk of traumatic stress exposure was obtained, which enabled a comparative analysis of neurobiological and emotional functioning.

The study protocol was approved by the Bioethics Committee of the Silesian Medical Chamber in Katowice, Poland (Decision No. 39/2018 of 10 October 2018, Annex No. 1 of 28 October 2019, Annex No. 2 of 19 June 2023, Annex No. 3 of 31 March 2025).

### 2.2. Brief-COPE Questionnaire

To assess coping strategies, the Brief-COPE questionnaire was used, a shortened version of the tool developed by Charles S. Carver (1997), designed to measure ways of responding to stressful situations [50]. This instrument consists of 28 items grouped into 14 two-item subscales, each reflecting a distinct coping strategy—either adaptive (e.g., planning, active coping, and seeking support) or maladaptive (e.g., avoidance, denial, use of psychoactive substances, and self-blame).

Participants responded to each statement by indicating the frequency of using a given strategy in stressful situations, using a four-point Likert scale: 1—I do not do this at all; 2—I do this a little bit; 3—I do this a moderate amount; and 4—I do this very often.

Scores for each subscale range from 2 to 8 points, as each strategy is assessed on the basis of two items. The higher the score in a given subscale, the more frequently the corresponding coping strategy is used.

In interpreting the results, it is assumed that: scores of 2–3 points indicate rare or incidental use of a given strategy; scores of 4–5 points suggest moderate frequency of its use; and scores of 6–8 points indicate frequent or very frequent use of the strategy when facing stress.

Analysis of Brief-COPE results can be conducted both qualitatively—by identifying dominant strategies—and quantitatively, for example, by calculating averages for groups of adaptive and maladaptive strategies. Adaptive strategies are generally associated with greater psychological resilience, more effective adjustment, and lower severity of psychopathological symptoms. Maladaptive strategies, such as avoidance, denial, or substance use, may contribute to the persistence of PTSD, depression, or anxiety symptoms and impair cognitive and emotional functioning.

### 2.3. Methodology of Biological Material Collection, Serum Preparation, and Biomarker Analysis

Venous blood was collected in the morning hours (between 7:30 and 9:30) under standardized conditions. Participants were fasting and had not taken psychotropic medications for at least 24 h prior to the study. Immediately after collection, samples were centrifuged, and the obtained serum was frozen and stored at −80 °C until analysis.

Concentrations of selected biomarkers—IGF-1, CASP-9, nNOS, and IL-10—were measured using Enzyme-Linked Immunosorbent Assay (ELISA) tests. Validated, commercially available diagnostic kits with Conformité Européenne/In Vitro Diagnostic (CE/IVD) certification were used. Each kit contained an appropriate calibration curve, negative control (blank), and a complete set of reagents required for the assay. Serum was diluted according to the manufacturer’s instructions.

Microtiter plates coated with specific capture antibodies were filled with prepared samples and standards. Incubation lasted 60 min at room temperature with orbital shaking (300 rpm) to facilitate antigen–antibody binding. The plates were then washed, biotinylated detection antibodies were added, and a further 60 min incubation under the same conditions was performed.

After another washing step, a streptavidin–Horseradish Peroxidase (HRP) complex was added, followed by a 30 min incubation. After the final wash, 100 µL of tetramethylbenzidine (TMB) substrate solution was added to each well. The color reaction proceeded for 10 min at room temperature and was stopped by adding stop solution (sulfuric acid). Absorbance was measured at 450 nm using a microplate reader. Analyte concentrations were calculated from the calibration curve using a Four-Parameter Logistic (4-PL) model.

Separate calibration curves were prepared for each parameter using recombinant protein standards. Analytical characteristics of the tests are listed below:

**IGF-1:** measurement range: 0.2–40 ng/mL; sensitivity: 0.153 ng/mL; manufacturer: (SunRedBio, Shanghai, China), catalog no.: 201-12-0104.

**CASP-9:** measurement range: 0.08–20 ng/mL; sensitivity: 0.073 ng/mL; manufacturer: (SunRedBio, Shanghai, China); catalog no.: 201-12-0969.

**nNOS:** measurement range: 0.3–70 ng/mL; sensitivity: 0.244 ng/mL; manufacturer: (SunRedBio, Shanghai, China); catalog no.: 201-12-5515.

**IL-10:** measurement range: 10–3000 pg/mL; sensitivity: 9.012 pg/mL; manufacturer: (SunRedBio, Shanghai, China); catalog no.: 201-12-0103.

### 2.4. Statistical Analysis

All statistical tests were two-tailed, with a significance level set at α = 0.05. Continuous variables, such as age, biomarker levels (IGF-1, CASP-9, nNOS, and IL-10), and Brief-COPE subscale scores (Problem-Focused Coping, Avoidant Coping, Emotion-Focused Coping, and Overall Coping), were assessed for normality using the Shapiro–Wilk test. Due to non-normal distributions, these variables are presented as medians with IQR and were compared across groups using the Kruskal–Wallis test, followed by Dunn’s post hoc test with Holm–Bonferroni correction for pairwise comparisons when significant differences were detected. The results of the post hoc tests are reported using the Compact Letter Display (CLD) approach, where different letters indicate significant differences between groups, and the same or overlapping letters denote no significant difference. Categorical variables, including marital status, education level, and sports performance/training frequency, are reported as frequencies and percentages and were analyzed using the Chi-square test or Fisher’s exact test when expected cell counts were less than 5. For significant overall tests, post hoc pairwise comparisons were conducted using Chi-square or Fisher’s exact test, with *p*-values adjusted using the False Discovery Rate (FDR) method. Correlations between biomarkers (IGF-1, CASP-9, nNOS, and IL-10) and Brief-COPE subscale scores were evaluated using Spearman’s rank correlation coefficient, given the non-parametric nature of the data. No correction for multiple comparisons was applied to the correlation analyses due to their exploratory nature [51]. The magnitude of the correlation coefficients was interpreted using the approach by Schober P. et al. [52], classifying absolute values as negligible (0.00–0.10), weak (0.10–0.39), moderate (0.40–0.69), strong (0.70–0.89), or very strong (0.90–1.00). Correlation analyses were conducted separately for each group to assess the influence of PTSD status and duration on these relationships.

### 2.5. Statistical Tool

Analyses were conducted using the R Statistical language (version 4.3.3; [53] R Core Team, 2024) on Windows 11 pro 64 bit (build 26100), using the packages ggpubr (version 0.6.0; [54]), performance (version 0.12.3; [55]), report (version 0.5.8; [56]), gtsummary (version 1.7.2; [57]), ggplot2 (version 3.5.0; [58]), readxl (version 1.4.3; [59]) and dplyr (version 1.1.4; [60]).

## 3. Results

### 3.1. Characteristics and Clinical Profiles of Male Mine Rescue Workers and Former Miners by PTSD Status

The results in Table 1 assessed differences in demographics, lifestyle, biomarkers, and coping strategies across groups.

Demographics: The overall median age was 34.0 years (IQR 28.8–41.0), with no significant difference across groups (*p* = 0.524). Marital status varied significantly (*p* < 0.001): the no-PTSD group had a higher proportion of married individuals (75.0%) compared to 27.3% (≤5 y) and 19.4% (>5 y) in the PTSD groups, while divorced or separated status was more common in PTSD groups (66.7% ≤ 5 y, 61.3% > 5 y) than controls (21.4%); pairwise adjusted *p*-values showed no difference between PTSD ≤ 5 y and >5 y (adjusted *p* = 0.285), but significant differences between PTSD ≤ 5 y vs. control (adjusted *p* < 0.001) and PTSD > 5 y vs. control (adjusted *p* < 0.001). Education levels (vocational vs. higher) showed no significant variation (*p* = 0.329), with roughly half the cohort having vocational training (52.2%).

Lifestyle: Sports performance and training frequency differed significantly (*p* = 0.006). Daily training was more prevalent in PTSD groups (66.7% ≤ 5 y, 74.2% > 5 y) than controls (39.3%), while weekly training (1 time/week) was exclusive to controls (17.9%), indicating a potential link between PTSD history and increased physical activity; pairwise adjusted *p*-values indicated no difference between PTSD ≤ 5 y and >5 y (adjusted *p* = 0.590), but significant differences between PTSD ≤ 5 y vs. control (adjusted *p* = 0.018) and PTSD > 5 y vs. control (adjusted *p* = 0.016).

Biomarkers: Biomarker profiles showed marked differences (all *p* < 0.001). IGF-1 levels were highest in controls (37.1 nmol/mL, IQR 32.5–44.6), followed by PTSD > 5 y (13.6 nmol/mL, IQR 7.6–19.6), and lowest in PTSD ≤ 5 y (4.9 nmol/mL, IQR 4.0–5.8), indicating a possible recovery trend with time since PTSD (Figure 1A); pairwise adjusted *p*-values were <0.001 for all comparisons (PTSD ≤ 5 y vs. >5 y, ≤5 y vs. control, >5 y vs. control). CASP-9 and nNOS levels were elevated in PTSD ≤ 5 y (23.8 ng/mL, IQR 20.8–26.5; 62.3 ng/mL, IQR 46.6–82.1) compared to PTSD > 5 y (6.8 ng/mL, IQR 3.7–16.6; 19.1 ng/mL, IQR 9.5–35.7) and controls (2.7 ng/mL, IQR 1.8–3.3; 5.7 ng/mL, IQR 4.8–7.1), revealing greater apoptotic and oxidative stress in recent PTSD (Figure 1B and Figure 1C, respectively); pairwise adjusted *p*-values were <0.001 for all comparisons. Conversely, IL-10, an anti-inflammatory marker, was highest in controls (3159.0 ng/L, IQR 2814.0–3516.2), intermediate in PTSD > 5 y (833.7 ng/L, IQR 523.0–1448.5), and lowest in PTSD ≤ 5 y (279.0 ng/L, IQR 231.0–345.3), reflecting reduced inflammation regulation in recent PTSD (Figure 1D); pairwise adjusted *p*-values were <0.001 for all comparisons.

Coping strategies: Coping strategies were assessed using the Brief-COPE inventory, with subscale scores reflecting distinct psychological responses to stress: Problem-Focused Coping, Avoidant Coping, Emotion-Focused Coping, and Overall Coping. Higher scores indicate greater use of each coping style. Significant differences were observed across all subscales (*p* < 0.001) between all groups.

The Problem-Focused Coping subscale measures active, solution-oriented strategies (e.g., planning and seeking support). The overall median score was 14.0 (IQR 13.0–28.0), representing moderate use relative to the 0–32 range. Controls exhibited a strikingly high median of 30.0 (IQR 29.0–30.0), approaching the scale’s maximum, indicating robust adaptive coping. In contrast, PTSD ≤ 5 y scored 13.0 (IQR 12.0–14.0) and PTSD > 5 y scored 14.0 (IQR 12.5–16.0), both in the lower half of the range, reflecting limited engagement in proactive strategies (Figure 2A). This disparity indicates that PTSD history, regardless of duration, impairs the ability to mobilize problem-focused coping, a critical deficit in psychiatric resilience; pairwise adjusted *p*-values showed no difference between PTSD ≤ 5 y and >5 y (adjusted *p* = 0.180), but significant differences between PTSD ≤ 5 y vs. control (adjusted *p* < 0.001) and PTSD > 5 y vs. control (adjusted *p* < 0.001).

The Avoidant Coping subscale captures maladaptive behaviors (e.g., denial and disengagement), with a range of 0–32. The overall median was 24.0 (IQR 13.8–25.0), indicating substantial reliance on avoidance across the cohort. PTSD ≤ 5 y showed the highest median at 26.0 (IQR 25.0–26.0), nearing the upper limit, followed by PTSD > 5 y at 24.0 (IQR 23.0–25.0), both reflecting pronounced avoidance. Controls scored markedly lower at 12.0 (IQR 11.0–13.0), within the lower third of the range, inferring minimal use of maladaptive strategies (Figure 2B). The elevated scores in PTSD groups align with psychiatric features of trauma-related avoidance, with recent PTSD (≤5 y) showing the most severe presentation; pairwise adjusted *p*-values were 0.004 for PTSD ≤ 5 y vs. >5 y, and <0.001 for both PTSD groups vs. control.

The Emotion-Focused Coping subscale assesses emotional regulation and support-seeking (e.g., acceptance and positive reframing), ranging from 0–48. The overall median was 27.5 (IQR 22.0–42.0), indicating moderate-to-high use. Controls scored 43.0 (IQR 42.0–44.0), near the upper bound, reflecting strong emotional resilience. PTSD ≤ 5 y scored 27.0 (IQR 24.0–30.0), in the mid-range, while PTSD > 5 y scored 21.0 (IQR 19.0–22.5), in the lower half, indicating a decline in emotional coping capacity with longer PTSD duration (Figure 2C). The No PTSD group’s high score contrasts with the PTSD groups’ reduced ability to leverage emotional strategies, a common psychiatric impairment in trauma survivors; pairwise adjusted *p*-values were <0.001 for all comparisons.

The total Overall Coping score (0–112) integrates all coping styles, with an overall median of 65.5 (IQR 60.8–82.0), slightly above the midpoint, indicating a balanced but variable coping repertoire. Controls scored 84.5 (IQR 83.0–86.0), in the upper quartile, reflecting comprehensive coping strength. PTSD ≤ 5 y scored 66.0 (IQR 62.0–68.0), near the median, while PTSD > 5 y scored 59.0 (IQR 56.0–62.0), below the midpoint, demonstrating a progressive reduction in overall coping capacity over time (Figure 2D). The PTSD groups’ scores, particularly >5 y, indicate a diminished global coping profile compared to controls; pairwise adjusted *p*-values were 0.001 for PTSD ≤ 5 y vs. >5 y, and <0.001 for both PTSD groups vs. control.

Clinical implications for coping strategies findings: PTSD significantly alters coping dynamics. Controls demonstrate a near-optimal profile, with high problem-focused (30.0/32) and emotion-focused (43.0/48) scores and low avoidance (12.0/32), consistent with psychiatric resilience in the absence of trauma sequelae. In contrast, PTSD ≤ 5 y patients exhibit a maladaptive pattern: low problem-focused coping (13.0/32), high avoidance (26.0/32), and moderate emotion-focused coping (27.0/48), reflecting acute post-traumatic stress with avoidance as a dominant feature—akin to DSM-5 PTSD Criterion C (avoidance symptoms). PTSD > 5 y patients show a slight shift, with marginally improved problem-focused coping (14.0/32) but reduced emotion-focused coping (21.0/48) and persistent avoidance (24.0/32), revealing a chronic adaptation where emotional regulation weakens over time.

Duration of PTSD exposure influences coping profiles. Within 5 years, the high avoidance (26.0/32) and low problem-focused scores (13.0/32) indicate an acute, defensively oriented response, potentially linked to active symptom burden. Beyond 5 years, avoidance remains elevated (24.0/32), but the drop in emotion-focused coping (21.0/48) and overall score (59.0/112) implies a fatigue or depletion of adaptive resources, a pattern observed in chronic PTSD where initial coping efforts may wane.

Coping symptoms do not intensify over time; rather, they shift. The PTSD ≤ 5 y group’s profile aligns with acute PTSD, with pronounced avoidance and limited adaptive coping. The PTSD > 5 y group shows a partial normalization (e.g., slight increase in problem-focused coping), but the decline in emotion-focused coping and overall score demonstrates a move toward emotional disengagement rather than recovery to control levels. This trajectory indicates that, in this cohort, PTSD symptoms may stabilize or adapt rather than resolve fully, with lingering psychiatric vulnerability.

Detailed pairwise adjusted *p*-values for all significant parameters are provided in Table 2.

### 3.2. Correlations Between Biomarkers and Coping Strategies in Male Mine Rescue Workers and Former Miners by PTSD Status

#### 3.2.1. Association Between IGF-1 Levels and Coping Scores

IGF-1 concentrations exhibit distinct associations with coping characteristics across groups. For Problem-Focused Coping (Figure 3A), the No PTSD group shows a negligible correlation (Rho = 0), Past PTSD > 5 y demonstrates a weak negative correlation (Rho = −0.29), and Past PTSD ≤ 5 y reveals a negligible positive correlation (Rho = 0.04). For Avoidant Coping (Figure 3B), correlations are weak or negligible, with No PTSD at Rho = −0.13 (weak), Past PTSD > 5 y at Rho = −0.01 (negligible), and Past PTSD ≤ 5 y at Rho = −0.12 (weak). Emotion-Focused Coping (Figure 3C) presents a weak positive correlation in No PTSD (Rho = 0.2) and Past PTSD > 5 y (Rho = 0.21), but a weak negative correlation in Past PTSD ≤ 5 y (Rho = −0.39). Overall scores (Figure 3D) follow a similar pattern, with No PTSD at Rho = 0.03 (negligible), Past PTSD > 5 y at Rho = −0.02 (negligible), and Past PTSD ≤ 5 y at Rho = −0.35 (weak). Clinically, IGF-1, a marker of growth and neuroprotection, exerts minimal influence on coping in healthy controls. However, in PTSD patients, higher IGF-1 levels impair active coping (Problem-Focused) in chronic cases (>5 y), while lower levels in acute cases (≤5 y) correlate with reduced emotional and overall coping, reflecting heightened stress responses and potential HPA axis dysregulation.

Reflection of physiological and psychological balance: A balanced physiological and psychological state is characterized by biomarker levels and coping scores indicative of stability, as observed in the No PTSD group. In this cohort, IGF-1 ranges from 16 to 57 nmol/mL, with coping scores consistently high (Problem-Focused: 28–32/32; Emotion-Focused: 40–45/48; and Overall: 80–89/112) and avoidance low (9–15/32), demonstrating a state where neuroprotection supports adaptive coping. In contrast, Past PTSD ≤ 5 y patients exhibit a disrupted equilibrium, with suppressed IGF-1 (0–10 nmol/mL), low adaptive coping (Problem-Focused: 10–16/32; Emotion-Focused: 20–33/48), and high avoidance (21–29/32), indicating the impact of acute trauma. Past PTSD > 5 y patients show an intermediate state, with IGF-1 (3–45 nmol/mL) approaching control levels and coping scores (Problem-Focused: 9–21/32; Emotion-Focused: 16–28/48) improving but remaining below controls, reflecting partial recovery over time.

Clinical effect of deviations from balance: Deviations from this balanced state in PTSD groups carry significant clinical implications. In Past PTSD ≤ 5 y, low IGF-1 (0–10 nmol/mL) indicates a suppressed neuroprotective state, likely driven by HPA axis dysregulation, correlating with impaired adaptive coping (e.g., Problem-Focused: 10–16/32) and elevated avoidance (21–29/32). This imbalance exacerbates psychiatric symptoms, including hyperarousal (DSM-5 Criterion E) and avoidance (Criterion C), potentially increasing the risk of chronic PTSD and comorbidities such as depression. In Past PTSD > 5 y, the negative correlation between IGF-1 and Problem-Focused Coping (Rho = −0.29; Figure 3A) demonstrates that rising IGF-1 levels (3–45 nmol/mL) do not enhance active coping, possibly reflecting learned helplessness or persistent negative alterations in cognition (Criterion D). Clinically, these deviations impair recovery, perpetuating maladaptive coping and diminishing resilience to future stressors.

Effect of PTSD status on associations: PTSD status markedly influences the associations between IGF-1 and coping strategies. The No PTSD group displays near-zero correlations across subscales (Rho = −0.13 to 0.2), indicating that IGF-1 levels do not impact coping in the absence of trauma, consistent with psychological resilience. Conversely, Past PTSD ≤ 5 y exhibits stronger negative correlations, particularly for Emotion-Focused (Rho = −0.39; Figure 3C) and Overall scores (Rho = −0.35; Figure 3D), demonstrating that low IGF-1 levels in the acute phase impair emotional regulation and overall coping capacity. Past PTSD > 5 y reveals a varied pattern, with a negative correlation for Problem-Focused Coping (Rho = −0.29; Figure 3A) but a positive correlation for Emotion-Focused Coping (Rho = 0.21; Figure 3C), highlighting that PTSD status alters both the direction and strength of associations, with acute PTSD intensifying negative relationships and chronic PTSD presenting a mixed profile.

Stability of coping subscales in PTSD patients: Among PTSD patients, coping subscales demonstrate varying stability in their associations with IGF-1. Avoidant Coping (Figure 3B) shows stable, weak or negligible negative correlations in both PTSD groups (≤5 y: Rho = −0.12, >5 y: Rho = −0.01), indicating that avoidance behaviors remain consistently high and minimally influenced by IGF-1 across PTSD duration. Problem-Focused Coping (Figure 3A) shifts from negligible positive correlation in ≤5 y (Rho = 0.04) to a weak negative correlation in >5 y (Rho = −0.29), reflecting instability as PTSD duration increases. Emotion-Focused (Figure 3C) and Overall scores (Figure 3D) are less stable, with Past PTSD ≤ 5 y showing weak negative correlations (Emotion-Focused: Rho = −0.39, Overall: Rho = −0.35) that transition to a weak positive correlation (Emotion-Focused: Rho = 0.21) or negligible correlation (Overall: Rho = −0.02) in >5 y, indicating dynamic changes over time.

Coping subscales most affected in PTSD patients: PTSD patients experience the most pronounced impact on Problem-Focused and Emotion-Focused Coping subscales. In Past PTSD ≤ 5 y, Problem-Focused scores (Figure 3A) are low (10–16/32) with negligible correlation (Rho = 0.04), and Emotion-Focused scores (Figure 3C) (20–33/48) exhibit a weak negative correlation (Rho = −0.39), indicating substantial impairment in active and emotional coping. In Past PTSD > 5 y, Problem-Focused Coping (Figure 3A) remains low (9–21/32) with a weak negative correlation (Rho = −0.29), while Emotion-Focused Coping (Figure 3C) improves slightly (16–28/48) and shifts to a weak positive correlation (Rho = 0.21). Avoidant Coping (Figure 3B) is also impacted, with high scores in both PTSD groups (≤5 y: 21–29/32, >5 y: 21–26/32), though correlations remain weak or negligible, reflecting a persistent but less dynamic effect.

Coping subscales dynamically changing over PTSD duration: Emotion-Focused and Overall Coping subscales exhibit the most dynamic changes over PTSD duration. In Past PTSD ≤ 5 y, Emotion-Focused Coping (Figure 3C) demonstrates a weak negative correlation (Rho = −0.39), reflecting impaired emotional regulation, which transitions to a weak positive correlation (Rho = 0.21) in >5 y, indicating improved emotional coping as IGF-1 levels increase. Overall Coping (Figure 3D) shifts from a weak negative correlation (Rho = −0.35) in ≤5 y to negligible correlation (Rho = −0.02) in >5 y, reflecting a stabilization of global coping capacity. Problem-Focused Coping (Figure 3A) also changes, moving from negligible positive correlation (Rho = 0.04) in ≤5 y to a weak negative correlation (Rho = −0.29) in >5 y, indicating a decline in active coping over time. Avoidant Coping (Figure 3B) remains relatively stable, with consistently high scores and weak or negligible correlations across both PTSD durations.

#### 3.2.2. Association Between CASP-9 Levels and Coping Scores

CASP-9 concentrations, a marker of apoptosis and stress, correlate variably with coping characteristics in PTSD patients. For Problem-Focused Coping (Figure 4A), the No PTSD group shows negligible correlation (Rho = 0), Past PTSD > 5 y exhibits a weak negative correlation (Rho = −0.29), and Past PTSD ≤ 5 y displays negligible positive correlation (Rho = 0.01). For Avoidant Coping (Figure 4B), correlations are negligible, with Past PTSD ≤ 5 y at Rho = 0 and Past PTSD > 5 y at Rho = −0.07 (No PTSD not specified, but consistent with overall negligible trends). Emotion-Focused Coping (Figure 4C) reveals a weak positive correlation in No PTSD (Rho = 0.2), a weak negative correlation in Past PTSD > 5 y (Rho = −0.24), and a weak negative correlation in Past PTSD ≤ 5 y (Rho = −0.26). Overall scores (Figure 4D) show a weak positive correlation in No PTSD (Rho = 0.38), a negligible correlation in Past PTSD > 5 y (Rho = −0.03), and a weak negative correlation in Past PTSD ≤ 5 y (Rho = −0.17). Clinically, elevated CASP-9 levels in PTSD patients, particularly in the acute phase (≤5 y), impair adaptive coping (Problem-Focused and Emotion-Focused) and overall coping capacity, reflecting heightened stress responses and potential exacerbation of psychiatric symptoms such as hyperarousal (DSM-5 Criterion E) and avoidance (Criterion C).

Reflection of physiological and psychological balance: A balanced state, as observed in the No PTSD group, is marked by low CASP-9 levels (0.08–4.95 ng/mL) and high coping scores (Problem-Focused: 28–32/32; Emotion-Focused: 40–45/48; and Overall: 80–89/112), with low avoidance (9–15/32), indicating minimal apoptotic stress and robust adaptive coping. In contrast, Past PTSD ≤ 5 y patients exhibit elevated CASP-9 (12.3–35.05 ng/mL), low adaptive coping (Problem-Focused: 10–16/32; Emotion-Focused: 20–33/48), and high avoidance (21–29/32), reflecting a disrupted equilibrium due to acute trauma. Past PTSD > 5 y patients show intermediate CASP-9 levels (2.0–26.91 ng/mL) and slightly improved coping (Problem-Focused: 9–21/32; Emotion-Focused: 16–28/48), indicating partial recovery but persistent deficits.

Clinical effect of deviations from balance: Deviations from this balanced state in PTSD groups carry significant clinical implications. In Past PTSD ≤ 5 y, high CASP-9 (12.3–35.05 ng/mL) indicates elevated apoptotic stress, likely driven by trauma-induced dysregulation, correlating with impaired adaptive coping (e.g., Problem-Focused: 10–16/32) and elevated avoidance (21–29/32). This imbalance exacerbates psychiatric symptoms, including hyperarousal (DSM-5 Criterion E) and avoidance (Criterion C), potentially increasing the risk of chronic PTSD and comorbidities such as anxiety or depression. In Past PTSD > 5 y, the negative correlation between CASP-9 and Problem-Focused Coping (Rho = −0.29; Figure 4A) demonstrates that despite decreasing CASP-9 levels (2.0–26.91 ng/mL), active coping remains impaired, possibly reflecting learned helplessness or persistent negative alterations in cognition (Criterion D). Clinically, these deviations impair recovery, perpetuating maladaptive coping and diminishing resilience to future stressors.

Effect of PTSD status on associations: PTSD status markedly influences the associations between CASP-9 and coping strategies. The No PTSD group displays positive correlations across subscales (Emotion-Focused: Rho = 0.2; Overall: Rho = 0.38), indicating that low CASP-9 levels support emotional and overall coping, consistent with psychological resilience. Conversely, Past PTSD ≤ 5 y exhibits negative correlations, particularly for Emotion-Focused (Rho = −0.26; Figure 4C) and Overall scores (Rho = −0.17; Figure 4D), demonstrating that elevated CASP-9 levels in the acute phase impair emotional regulation and overall coping capacity. Past PTSD > 5 y reveals a varied pattern, with negative correlations for Problem-Focused (Rho = −0.29; Figure 4A) and Emotion-Focused Coping (Rho = −0.24; Figure 4C), highlighting that PTSD status alters both the direction and strength of associations, with acute PTSD intensifying negative relationships and chronic PTSD presenting a mixed profile.

Stability of coping subscales in PTSD patients: Among PTSD patients, coping subscales demonstrate varying stability in their associations with CASP-9. Avoidant Coping (Figure 4B) shows stable, negligible correlations in both PTSD groups (≤5 y: Rho = 0, >5 y: Rho = −0.07), indicating that avoidance behaviors remain consistently high and minimally influenced by CASP-9 across PTSD duration. Problem-Focused Coping (Figure 4A) shifts from a negligible positive correlation in ≤5 y (Rho = 0.01) to a weak negative correlation in >5 y (Rho = −0.29), reflecting instability as PTSD duration increases. Emotion-Focused (Figure 4C) and Overall scores (Figure 4D) are less stable, with Past PTSD ≤ 5 y showing weak negative correlations (Emotion-Focused: Rho = −0.26, Overall: Rho = −0.17) that remain weak negative (Emotion-Focused: Rho = −0.24) or transition to negligible (Overall: Rho = −0.03) in >5 y, indicating dynamic changes over time.

Coping subscales most affected in PTSD patients: PTSD patients experience the most pronounced impact on Problem-Focused and Emotion-Focused Coping subscales. In Past PTSD ≤ 5 y, Problem-Focused scores (Figure 4A) are low (10–16/32) with negligible correlation (Rho = 0.01), and Emotion-Focused scores (Figure 4C) (20–33/48) exhibit a weak negative correlation (Rho = −0.26), indicating substantial impairment in active and emotional coping. In Past PTSD > 5 y, Problem-Focused Coping (Figure 4A) remains low (9–21/32) with a weak negative correlation (Rho = −0.29), while Emotion-Focused Coping (Figure 4C) improves slightly (16–28/48) and shows a weak negative correlation (Rho = −0.24). Avoidant Coping (Figure 4B) is also impacted, with high scores in both PTSD groups (≤5 y: 21–29/32, >5 y: 21–26/32), though correlations remain negligible, reflecting a persistent but less dynamic effect.

Coping subscales dynamically changing over PTSD duration: Problem-Focused and Emotion-Focused Coping subscales exhibit the most dynamic changes over PTSD duration. In Past PTSD ≤ 5 y, Problem-Focused Coping (Figure 4A) demonstrates a negligible positive correlation (Rho = 0.01), reflecting minimal influence in the acute phase, which transitions to a weak negative correlation (Rho = −0.29) in >5 y, indicating a decline in active coping as duration increases. Emotion-Focused Coping (Figure 4C) shifts from a weak negative correlation (Rho = −0.26) in ≤5 y to a weaker negative correlation (Rho = −0.24) in >5 y, reflecting persistent impairment in emotional regulation over time. Overall Coping (Figure 4D) changes from a weak negative correlation (Rho = −0.17) in ≤5 y to negligible correlation (Rho = −0.03) in >5 y, suggesting stabilization of global coping capacity. Avoidant Coping (Figure 4B) remains relatively stable, with consistently high scores and negligible correlations across both PTSD durations.

#### 3.2.3. Association Between nNOS Levels and Coping Scores

nNOS concentrations, a marker of oxidative stress, correlate variably with coping characteristics in PTSD patients. For Problem-Focused Coping (Figure 5A), the No PTSD group shows negligible negative correlation (Rho = −0.08), Past PTSD > 5 y exhibits a weak negative correlation (Rho = −0.3), and Past PTSD ≤ 5 y displays a weak positive correlation (Rho = 0.19). For Avoidant Coping (Figure 5B), correlations are weak or negligible, with Past PTSD ≤ 5 y at Rho = −0.18 (weak) and Past PTSD > 5 y at Rho = 0 (negligible) (No PTSD not specified, but consistent with overall negligible trends). Emotion-Focused Coping (Figure 5C) reveals a weak negative correlation across all groups: No PTSD (Rho = −0.14), Past PTSD > 5 y (Rho = −0.22), and Past PTSD ≤ 5 y (Rho = −0.25). Overall scores (Figure 5D) show a weak negative correlation in Past PTSD ≤ 5 y (Rho = −0.15), a negligible correlation in Past PTSD > 5 y (Rho = 0.01), and a negligible correlation in No PTSD (Rho = −0.03). Clinically, elevated nNOS levels in PTSD patients, particularly in the chronic phase (>5 y), impair adaptive coping (Problem-Focused and Emotion-Focused) and overall coping capacity, reflecting persistent oxidative stress and potential exacerbation of psychiatric symptoms such as hyperarousal (DSM-5 Criterion E) and negative alterations in cognition (Criterion D).

Reflection of physiological and psychological balance: A balanced state, as observed in the No PTSD group, is marked by low nNOS levels (0.03–10.27 ng/mL) and high coping scores (Problem-Focused: 28–32/32; Emotion-Focused: 40–45/48; and Overall: 80–89/112), with low avoidance (9–15/32), indicating minimal oxidative stress and robust adaptive coping. In contrast, Past PTSD ≤ 5 y patients exhibit elevated nNOS (29.18–134.7 ng/mL), low adaptive coping (Problem-Focused: 10–16/32; Emotion-Focused: 20–33/48), and high avoidance (21–29/32), reflecting a disrupted equilibrium due to acute trauma. Past PTSD > 5 y patients show intermediate nNOS levels (5.42–66.2 ng/mL) and slightly improved coping (Problem-Focused: 9–21/32; Emotion-Focused: 16–28/48), indicating partial recovery but persistent deficits.

Clinical effect of deviations from balance: Deviations from this balanced state in PTSD groups carry significant clinical implications. In Past PTSD ≤ 5 y, high nNOS (29.18–134.7 ng/mL) indicates elevated oxidative stress, likely driven by trauma-induced dysregulation, correlating with impaired adaptive coping (e.g., Problem-Focused: 10–16/32) and elevated avoidance (21–29/32). This imbalance exacerbates psychiatric symptoms, including hyperarousal (DSM-5 Criterion E) and avoidance (Criterion C), potentially increasing the risk of chronic PTSD and comorbidities such as anxiety or depression. In Past PTSD > 5 y, the negative correlation between nNOS and Problem-Focused Coping (Rho = −0.3; Figure 5A) demonstrates that despite decreasing nNOS levels (5.42–66.2 ng/mL), active coping remains impaired, possibly reflecting learned helplessness or persistent negative alterations in cognition (Criterion D). Clinically, these deviations impair recovery, perpetuating maladaptive coping and diminishing resilience to future stressors.

Effect of PTSD status on associations: PTSD status markedly influences the associations between nNOS and coping strategies. The No PTSD group displays weak or negligible correlations across subscales (Rho = −0.14 to −0.03), indicating that low nNOS levels do not significantly impact coping in the absence of trauma, consistent with psychological resilience. Conversely, Past PTSD ≤ 5 y exhibits negative correlations, particularly for Emotion-Focused (Rho = −0.25; Figure 5C) and Overall scores (Rho = −0.15; Figure 5D), demonstrating that elevated nNOS levels in the acute phase impair emotional regulation and overall coping capacity. Past PTSD > 5 y reveals a varied pattern, with a negative correlation for Problem-Focused Coping (Rho = −0.3; Figure 5A) and Emotion-Focused Coping (Rho = −0.22; Figure 5C), highlighting that PTSD status alters both the direction and strength of associations, with chronic PTSD intensifying negative relationships for active coping and acute PTSD affecting emotional coping.

Stability of coping subscales in PTSD patients: Among PTSD patients, coping subscales demonstrate varying stability in their associations with nNOS. Avoidant Coping (Figure 5B) shows stable, weak or negligible correlations in both PTSD groups (≤5 y: Rho = −0.18, >5 y: Rho = 0), indicating that avoidance behaviors remain consistently high and minimally influenced by nNOS across PTSD duration. Problem-Focused Coping (Figure 5A) shifts from weak positive correlation in ≤5 y (Rho = 0.19) to a weak negative correlation in >5 y (Rho = −0.3), reflecting instability as PTSD duration increases. Emotion-Focused (Figure 5C) and Overall scores (Figure 5D) are less stable, with Past PTSD ≤ 5 y showing weak negative correlations (Emotion-Focused: Rho = −0.25; Overall: Rho = −0.15) that remain weak negative (Emotion-Focused: Rho = −0.22) or transition to negligible (Overall: Rho = 0.01) in >5 y, indicating dynamic changes over time.

Coping subscales most affected in PTSD patients: PTSD patients experience the most pronounced impact on Problem-Focused and Emotion-Focused Coping subscales. In Past PTSD ≤ 5 y, Problem-Focused scores (Figure 5A) are low (10–16/32) with a weak positive correlation (Rho = 0.19), and Emotion-Focused scores (Figure 5C) (20–33/48) exhibit a weak negative correlation (Rho = −0.25), indicating substantial impairment in active and emotional coping. In Past PTSD > 5 y, Problem-Focused Coping (Figure 5A) remains low (9–21/32) with a weak negative correlation (Rho = −0.3), while Emotion-Focused Coping (Figure 5C) improves slightly (16–28/48) and shows a weak negative correlation (Rho = −0.22). Avoidant Coping (Figure 5B) is also impacted, with high scores in both PTSD groups (≤5 y: 21–29/32, >5 y: 21–26/32), though correlations remain weak or negligible, reflecting a persistent but less dynamic effect.

Coping subscales dynamically changing over PTSD duration: Problem-Focused and Emotion-Focused Coping subscales exhibit the most dynamic changes over PTSD duration. In Past PTSD ≤ 5 y, Problem-Focused Coping (Figure 5A) demonstrates a weak positive correlation (Rho = 0.19), reflecting minimal influence in the acute phase, which transitions to a weak negative correlation (Rho = −0.3) in >5 y, indicating a decline in active coping as duration increases. Emotion-Focused Coping (Figure 5C) shifts from a weak negative correlation (Rho = −0.25) in ≤5 y to a weaker negative correlation (Rho = −0.22) in >5 y, reflecting persistent impairment in emotional regulation over time. Overall Coping (Figure 5D) changes from a weak negative correlation (Rho = −0.15) in ≤5 y to a negligible correlation (Rho = 0.01) in >5 y, indicating stabilization of global coping capacity. Avoidant Coping (Figure 5B) remains relatively stable, with consistently high scores and weak or negligible correlations across both PTSD durations.

#### 3.2.4. Association Between IL-10 Levels and Coping Scores

Concentrations of IL-10, an anti-inflammatory cytokine, correlate variably with coping characteristics in PTSD patients. For Problem-Focused Coping (Figure 6A), the No PTSD group shows a weak negative correlation (Rho = −0.13), Past PTSD > 5 y exhibits a weak negative correlation (Rho = −0.29), and Past PTSD ≤ 5 y displays a weak positive correlation (Rho = 0.11). For Avoidant Coping (Figure 6B), correlations are negligible or weak, with Past PTSD ≤ 5 y at Rho = −0.06 (negligible) and Past PTSD > 5 y at Rho = −0.13 (weak) (No PTSD not specified, but consistent with overall negligible trends). Emotion-Focused Coping (Figure 6C) reveals a weak positive correlation in No PTSD (Rho = 0.12), a weak negative correlation in Past PTSD > 5 y (Rho = −0.27), and a weak negative correlation in Past PTSD ≤ 5 y (Rho = −0.19). Overall scores (Figure 6D) show a weak negative correlation in Past PTSD ≤ 5 y (Rho = −0.2), a weak negative correlation in Past PTSD > 5 y (Rho = −0.12), and a negligible correlation in No PTSD (Rho = 0.08). Clinically, low IL-10 levels in PTSD patients, particularly in the acute phase (≤5 y), impair adaptive coping (Problem-Focused and Emotion-Focused) and overall coping capacity, reflecting heightened inflammatory responses and potential exacerbation of psychiatric symptoms such as hyperarousal (DSM-5 Criterion E) and avoidance (Criterion C).

Reflection of physiological and psychological balance: A balanced state, as observed in the No PTSD group, is marked by high IL-10 levels (2125–3884 ng/L) and high coping scores (Problem-Focused: 28–32/32; Emotion-Focused: 40–45/48; and Overall: 80–89/112), with low avoidance (9–15/32), indicating robust anti-inflammatory activity and adaptive coping. In contrast, Past PTSD ≤ 5 y patients exhibit suppressed IL-10 (10–491.82 ng/L), low adaptive coping (Problem-Focused: 10–16/32; Emotion-Focused: 20–33/48), and high avoidance (21–29/32), reflecting a disrupted equilibrium due to acute trauma. Past PTSD > 5 y patients show intermediate IL-10 levels (291.25–2157 ng/L) and slightly improved coping (Problem-Focused: 9–21/32; Emotion-Focused: 16–28/48), indicating partial recovery but persistent deficits.

Clinical effect of deviations from balance: Deviations from this balanced state in PTSD groups carry significant clinical implications. In Past PTSD ≤ 5 y, low IL-10 (10–491.82 ng/L) indicates suppressed anti-inflammatory activity, likely driven by trauma-induced dysregulation, correlating with impaired adaptive coping (e.g., Problem-Focused: 10–16/32) and elevated avoidance (21–29/32). This imbalance exacerbates psychiatric symptoms, including hyperarousal (DSM-5 Criterion E) and avoidance (Criterion C), potentially increasing the risk of chronic PTSD and comorbidities such as depression. In Past PTSD > 5 y, the negative correlation between IL-10 and Problem-Focused Coping (Rho = −0.29; Figure 6A) demonstrates that despite increasing IL-10 levels (291.25–2157 ng/L), active coping remains impaired, possibly reflecting learned helplessness or persistent negative alterations in cognition (Criterion D). Clinically, these deviations impair recovery, perpetuating maladaptive coping and diminishing resilience to future stressors.

Effect of PTSD status on associations: PTSD status markedly influences the associations between IL-10 and coping strategies. The No PTSD group displays weak or negligible correlations across subscales (Rho = −0.13 to 0.12), indicating that high IL-10 levels do not significantly impact coping in the absence of trauma, consistent with psychological resilience. Conversely, Past PTSD ≤ 5 y exhibits negative correlations, particularly for Emotion-Focused (Rho = −0.19; Figure 6C) and Overall scores (Rho = −0.2; Figure 6D), demonstrating that low IL-10 levels in the acute phase impair emotional regulation and overall coping capacity. Past PTSD > 5 y reveals a varied pattern, with negative correlations for Problem-Focused (Rho = −0.29; Figure 6A) and Emotion-Focused Coping (Rho = −0.27; Figure 6C), highlighting that PTSD status alters both the direction and strength of associations, with acute PTSD intensifying negative relationships for emotional and overall coping and chronic PTSD affecting active and emotional coping.

Stability of coping subscales in PTSD patients: Among PTSD patients, coping subscales demonstrate varying stability in their associations with IL-10. Avoidant Coping (Figure 6B) shows stable, negligible or weak negative correlations in both PTSD groups (≤5 y: Rho = −0.06, >5 y: Rho = −0.13), indicating that avoidance behaviors remain consistently high and minimally influenced by IL-10 across PTSD duration. Problem-Focused Coping (Figure 6A) shifts from a weak positive correlation in ≤5 y (Rho = 0.11) to a weak negative correlation in >5 y (Rho = −0.29), reflecting instability as PTSD duration increases. Emotion-Focused (Figure 6C) and Overall scores (Figure 6D) are less stable, with Past PTSD ≤ 5 y showing weak negative correlations (Emotion-Focused: Rho = −0.19; Overall: Rho = −0.2) that transition to weak negative (Emotion-Focused: Rho = −0.27) or weak negative (Overall: Rho = −0.12) in >5 y, indicating dynamic changes over time.

Coping subscales most affected in PTSD patients: PTSD patients experience the most pronounced impact on Problem-Focused and Emotion-Focused Coping subscales. In Past PTSD ≤ 5 y, Problem-Focused scores (Figure 6A) are low (10–16/32) with a weak positive correlation (Rho = 0.11), and Emotion-Focused scores (Figure 6C) (20–33/48) exhibit a weak negative correlation (Rho = −0.19), indicating substantial impairment in active and emotional coping. In Past PTSD > 5 y, Problem-Focused Coping (Figure 6A) remains low (9–21/32) with a weak negative correlation (Rho = −0.29), while Emotion-Focused Coping (Figure 6C) improves slightly (16–28/48) and shows a weak negative correlation (Rho = −0.27). Avoidant Coping (Figure 6B) is also impacted, with high scores in both PTSD groups (≤5 y: 21–29/32, >5 y: 21–26/32), though correlations remain negligible or weak, reflecting a persistent but less dynamic effect.

Coping subscales dynamically changing over PTSD duration: Problem-Focused and Emotion-Focused Coping subscales exhibit the most dynamic changes over PTSD duration. In Past PTSD ≤ 5 y, Problem-Focused Coping (Figure 6A) demonstrates a weak positive correlation (Rho = 0.11), reflecting minimal influence in the acute phase, which transitions to a weak negative correlation (Rho = −0.29) in >5 y, indicating a decline in active coping as duration increases. Emotion-Focused Coping (Figure 6C) shifts from a weak negative correlation (Rho = −0.19) in ≤5 y to a weak negative correlation (Rho = −0.27) in >5 y, reflecting worsening impairment in emotional regulation over time. Overall Coping (Figure 6D) changes from a weak negative correlation (Rho = −0.2) in ≤5 y to a weak negative correlation (Rho = −0.12) in >5 y, suggesting slight stabilization of global coping capacity. Avoidant Coping (Figure 6B) remains relatively stable, with consistently high scores and negligible or weak correlations across both PTSD durations.

## 4. Discussion

The present study demonstrates persistent neurobiological and emotional dysfunctions in men professionally exposed to prolonged traumatic stress. These alterations encompass dysregulation of the neuroendocrine–immune axis and the predominance of maladaptive coping strategies. The chronicity of posttraumatic stress disorder (PTSD) modifies these interactions—after more than five years, some biomarkers show partial normalization; however, this biological compensation is not accompanied by improvements in emotional functioning, indicating lasting neuroadaptive and behavioral deficits [61,62,63,64].

Reduced insulin-like growth factor 1 (IGF-1) activity appears central to the limited neuroplasticity observed in PTSD. IGF-1 supports neuronal survival, synaptogenesis, and neurogenesis in key regulatory regions such as the hippocampus and prefrontal cortex [65,66]. Experimental models have shown that IGF-1 supplementation mitigates oxidative stress, inhibits apoptosis, and upregulates neuroprotective genes [67,68]. Clinically, low IGF-1 correlates with greater PTSD severity and decreased FKBP5 expression—a regulator of glucocorticoid receptor sensitivity—suggesting intensified HPA axis dysregulation [69,70]. The modest rise in IGF-1 in chronic PTSD likely reflects a compensatory mechanism rather than full recovery, since it does not translate into improved coping or cognitive flexibility. These findings reinforce the concept of incomplete neurobiological restitution in the context of persistent psychological rigidity [71].

Caspase-9 (CASP-9), a key enzyme of the mitochondrial apoptotic cascade, represents another critical element in PTSD pathophysiology. Chronic stress and glucocorticoid overload increase mitochondrial membrane permeability, triggering cytochrome c release and activating caspase-dependent cell death [72,73,74]. Elevated CASP-9 levels indicate sustained neurodegenerative signaling, particularly within hippocampal, amygdalar, and prefrontal networks essential for emotional regulation and executive control. Even in chronic PTSD, partial reductions in CASP-9 remain above normative levels, implying ongoing mitochondrial vulnerability [75]. Beyond apoptosis, caspases influence synaptic remodeling and plasticity, linking molecular degeneration to behavioral dysfunction [76]. This mechanism may underlie the observed association between CASP-9 activity and diminished emotional or problem-focused coping.

Increased neuronal nitric oxide synthase (nNOS) activity further highlights the role of oxidative–nitrosative stress in PTSD. Although nitric oxide (NO) is vital for neurotransmission and plasticity, its overproduction generates reactive nitrogen species that damage neuronal lipids, proteins, and DNA [77,78]. This redox imbalance disrupts prefrontal–amygdalar connectivity, impairing emotional regulation, cognitive appraisal, and impulse control [79]. Neuroimaging studies confirm weakened functional coupling between these regions in PTSD [80]. The persistence of elevated nNOS, even in chronic cases, suggests a biochemical substrate of sustained vulnerability despite partial symptom remission.

Deficiency of interleukin-10 (IL-10), an anti-inflammatory cytokine, underscores the immunological dimension of PTSD. IL-10 regulates microglial activity and limits excessive neuroinflammation; its reduction contributes to prolonged immune activation and neuronal stress [81,82]. Low IL-10 has been linked to anxiety, depressive symptoms, poor sleep, and somatic morbidity [83]. The positive relationship between IL-10 and emotion-focused coping suggests that this cytokine may play a dual role—both immunological and affective—in restoring homeostasis [84]. In chronic PTSD, IL-10 increases modestly but remains below reference values, indicating incomplete recovery of immune regulation and persistent proinflammatory tone.

Integrating these observations reveals a reciprocal feedback loop between biological and psychological dysfunctions. Reduced neurotrophic support (IGF-1), heightened apoptotic and oxidative signaling (CASP-9 and nNOS), and deficient anti-inflammatory control (IL-10) jointly foster neuronal instability and emotional dysregulation. These neurobiological abnormalities parallel a behavioral profile dominated by avoidance and reduced use of adaptive coping strategies such as planning, problem-solving, or social support. Avoidance, although initially protective, prevents effective emotional processing of trauma and perpetuates the chronicity of PTSD [85,86,87,88,89]. Consequently, maladaptive coping may exacerbate oxidative–inflammatory stress, while biological dysregulation further limits psychological flexibility [90,91].

These results highlight the need for early, integrated therapeutic intervention targeting both neurobiological and psychological domains. On the biological level, strategies enhancing IGF-1 signaling, modulating nitric oxide pathways, and restoring cytokine balance could promote neuroplasticity and counteract mitochondrial dysfunction. On the psychological level, interventions aimed at restructuring avoidance and reinforcing adaptive coping—through cognitive–behavioral therapy, mindfulness, or emotion regulation training—may strengthen prefrontal control over limbic hyperactivity. Combined approaches addressing neuroimmune mechanisms and coping processes may therefore yield synergistic effects, where biological stabilization enhances therapeutic engagement, and improved coping, in turn, normalizes stress-related neurochemical cascades.

Professionals continuously exposed to life-threatening environments, such as rescue or military personnel, constitute a particularly vulnerable group. Chronic activation of neuroendocrine and immune pathways may entrench maladaptive stress responses, emphasizing the importance of periodic biomarker monitoring and preventive interventions fostering resilience and recovery.

In summary, the current findings support a multidimensional model of PTSD in which neurobiological and behavioral mechanisms interact dynamically. Even when partial normalization of biomarkers occurs, emotional and cognitive deficits persist, indicating that full recovery requires synchronization between neural, immune, and psychological systems. Personalized interventions that integrate biological modulation and psychotherapy may thus represent the most promising avenue for restoring both neurobiological stability and adaptive functioning in PTSD [92].

Several limitations must be acknowledged regarding the correlation analyses in this study. First, the exploratory approach, without adjustment for multiple comparisons, increases the risk of Type I errors, particularly given the modest sample size (N = 92) and subgroup divisions, which may have resulted in underpowered detections of weaker associations. Consequently, many correlations were non-significant or of negligible to weak magnitude, potentially limiting the generalizability and robustness of the findings. Additionally, the cross-sectional design precludes inferences about causality, as the observed relationships between biomarkers and coping strategies could be bidirectional or influenced by unmeasured confounders such as comorbid conditions, medication use, or lifestyle factors. The study population, restricted to adult men with trauma histories, further constrains applicability to broader demographics, including women or diverse age groups. Future research should validate these results in larger, longitudinal cohorts to elucidate temporal dynamics and causal pathways, while incorporating advanced statistical controls to enhance reliability.

## 5. Conclusions

The conducted study confirms that chronic post-traumatic stress disorder is associated with persistent neurobiological and emotional disturbances, which may occur even despite partial alleviation of clinical symptoms. Serum biomarker analysis revealed significant alterations in areas related to neuroplasticity (IGF-1), mitochondrial apoptosis (CASP-9), oxidative–nitrosative stress (nNOS), and immune regulation (IL-10). These disturbances were closely linked to the coping strategy profile—regardless of PTSD duration, avoidant strategies predominated, while deficits in adaptive strategies correlated with elevated levels of neurotoxic and pro-apoptotic markers. These findings support the hypothesis of a bidirectional feedback loop between psychological mechanisms and the physiological stress response. Although in the group with PTSD lasting more than five years a partial normalization of some biological markers was observed, the persistence of maladaptive behavioral patterns suggests long-term impairments in emotion regulation and cognitive processes. This underscores the need for comprehensive therapeutic interventions that account for both biological and psychological mechanisms of the disorder—including anti-inflammatory support, modulation of the stress response, and individualized cognitive-behavioral approaches. Future studies are recommended to include longitudinal analyses, a broader demographic spectrum, and an expanded molecular panel, which will allow for a more complete understanding of the processes underlying chronic PTSD and the possibilities for its treatment.

## Figures and Tables

**Figure 1 cimb-47-00868-f001:**
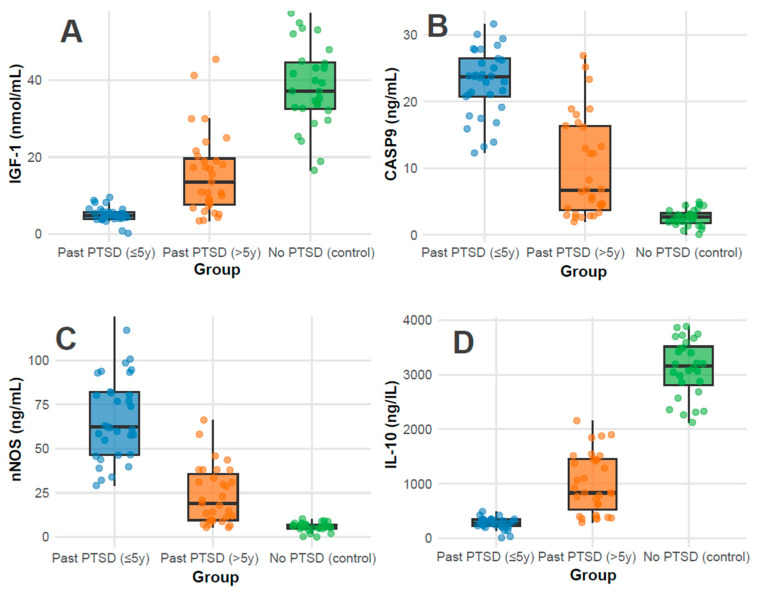
Distributions of Serum Biomarker Levels of IGF-1 (**A**), CASP-9 (**B**), nNOS (**C**), and IL-10 (**D**) across groups.

**Figure 2 cimb-47-00868-f002:**
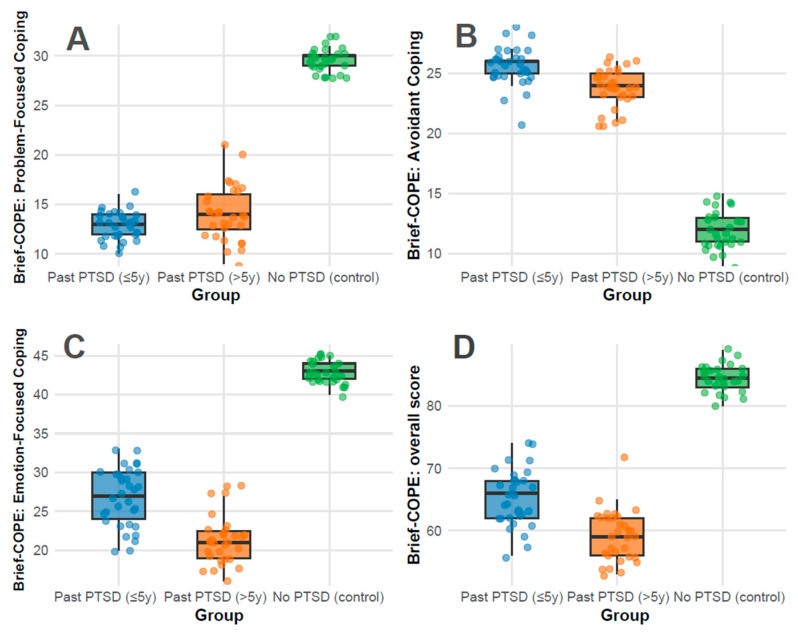
Distributions of Brief-COPE subscale scores: Problem-Focused Coping (**A**), Avoidant Coping (**B**), Emotion-Focused Coping (**C**), and Overall Score (**D**) across groups.

**Figure 3 cimb-47-00868-f003:**
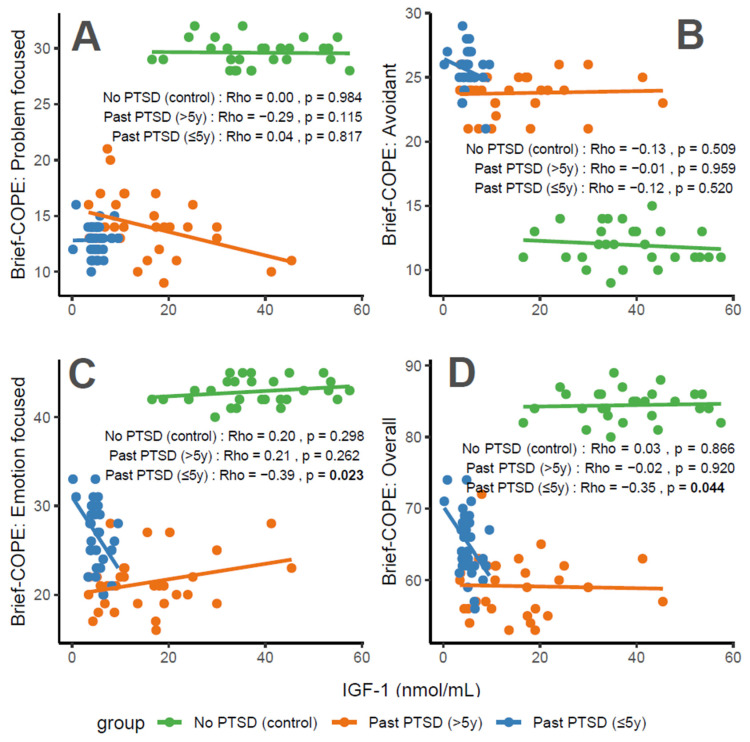
Correlations between IGF-1 biomarker and Brief COPE scores: problem-focused coping (**A**), avoidant coping (**B**), emotion-focused coping (**C**), and overall score (**D**) by PTSD status.

**Figure 4 cimb-47-00868-f004:**
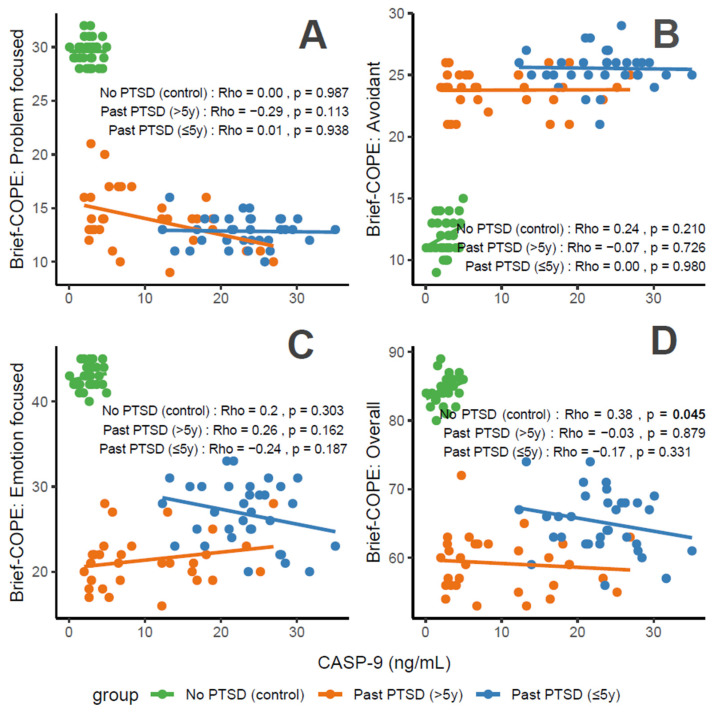
Correlations between CASP-9 biomarker and Brief COPE scores: problem-focused coping (**A**), avoidant coping (**B**), emotion-focused coping (**C**), and overall score (**D**) by PTSD status.

**Figure 5 cimb-47-00868-f005:**
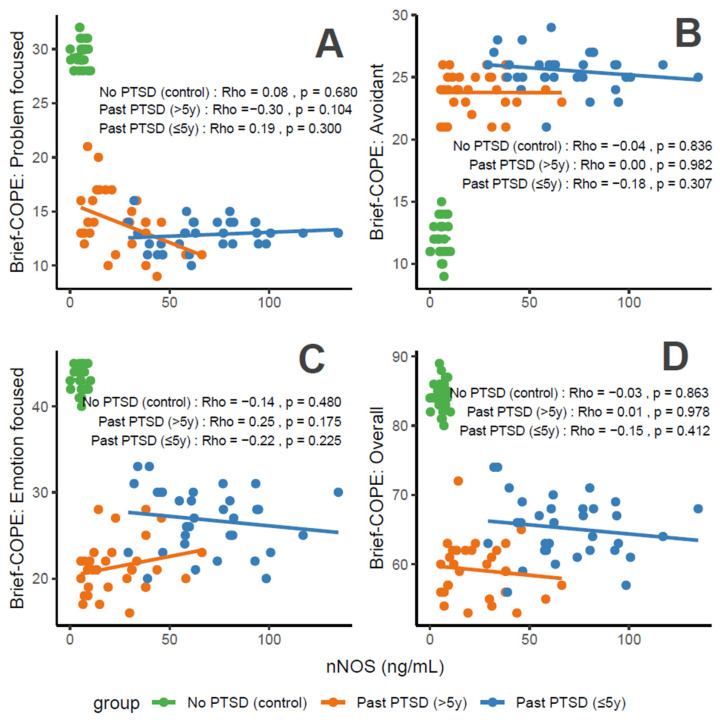
Correlations between nNOS biomarker Brief COPE scores: problem-focused coping (**A**), avoidant coping (**B**), emotion-focused coping (**C**), and overall score (**D**) by PTSD status.

**Figure 6 cimb-47-00868-f006:**
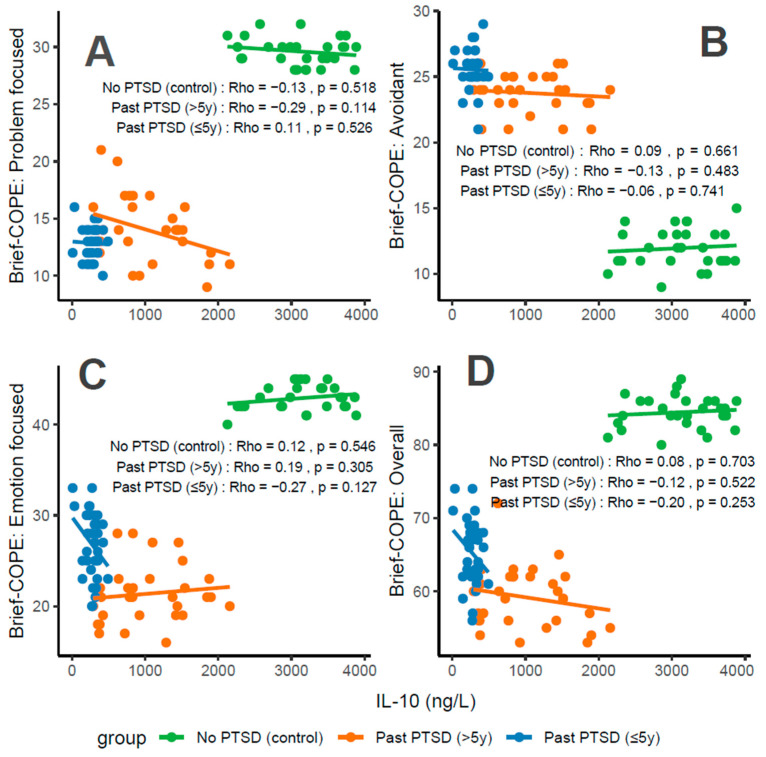
Correlations between IL-10 biomarker Brief COPE scores: problem-focused coping (**A**), avoidant coping (**B**), emotion-focused coping (**C**), and overall score (**D**) by PTSD status.

**Table 1 cimb-47-00868-t001:** Demographic, lifestyle, biomarker, and coping characteristics of male mine rescue workers and former miners with a history of life-threatening incidents by PTSD status (N = 92).

Characteristic	Overall (N = 92)	Past PTSD (≤5 y) (N = 33)	Past PTSD (>5 y) (N = 31)	No PTSD (Control) (N = 28)	*p*-Value
Demographics					
Age, median (IQR)	34.0 (28.8, 41.0)	34.0 (31.0, 41.0)	36.0 (29.5, 41.0)	33.5 (24.2, 41.5)	0.524
Marital status, n (%):					<0.001
divorced or separated	47 (51.1%)	22 (66.7%) ^A^	19 (61.3%) ^A^	6 (21.4%) ^B^	0.001
married	36 (39.1%)	9 (27.3%) ^B^	6 (19.4%) ^B^	21 (75.0%) ^A^	<0.001
never married	9 (9.8%)	2 (6.1%)	6 (19.4%)	1 (3.6%)	
Education, n (%):					0.339
vocational	48 (52.2%)	14 (42.4%)	17 (54.8%)	17 (60.7%)	
higher	44 (47.8%)	19 (57.6%)	14 (45.2%)	11 (39.3%)	
Lifestyle					
Sports performance and training, n (%):					0.006
daily	56 (60.9%)	22 (66.7%) ^A^	23 (74.2%) ^A^	11 (39.3%) ^B^	0.016
1 time in a week	5 (5.4%)	0 (0.0%) ^B^	0 (0.0%) ^B^	5 (17.9%) ^A^	0.002
2–3 times in a week	31 (33.7%)	11 (33.3%)	8 (25.8%)	12 (42.9%)	0.383
Biomarkers, median (IQR)					
IGF-1 (nmol/mL)	10.8 (5.2, 32.3)	4.9 (4.0, 5.8) ^C^	13.6 (7.6, 19.6) ^B^	37.1 (32.5, 44.6) ^A^	<0.001
CASP-9 (ng/mL)	10.2 (3.0, 22.9)	23.8 (20.8, 26.5) ^A^	6.8 (3.7, 16.6) ^B^	2.7 (1.8, 3.3) ^C^	<0.001
nNOS (ng/mL)	25.7 (7.0, 57.9)	62.3 (46.6, 82.1) ^A^	19.1 (9.5, 35.7) ^B^	5.7 (4.8, 7.1) ^C^	<0.001
IL-10 (ng/L)	806.9 (319.2, 2411.8)	279.0 (231.0, 345.3) ^C^	833.7 (523.0, 1448.5) ^B^	3159.0 (2814.0, 3516.2) ^A^	<0.001
Coping strategies (Brief-COPE), median (IQR) score					
Problem-focused coping [0–32 score]	14.0 (13.0, 28.0)	13.0 (12.0, 14.0) ^B^	14.0 (12.5, 16.0) ^B^	30.0 (29.0, 30.0) ^A^	<0.001
Avoidant coping [0–32 score]	24.0 (13.8, 25.0)	26.0 (25.0, 26.0) ^A^	24.0 (23.0, 25.0) ^B^	12.0 (11.0, 13.0) ^C^	<0.001
Emotion-focused coping [0–48 score]	27.5 (22.0, 42.0)	27.0 (24.0, 30.0) ^B^	21.0 (19.0, 22.5) ^C^	43.0 (42.0, 44.0) ^A^	<0.001
Overall score [0–112 score]	65.5 (60.8, 82.0)	66.0 (62.0, 68.0) ^B^	59.0 (56.0, 62.0) ^C^	84.5 (83.0, 86.0) ^A^	<0.001

Notes: The results of post hoc tests, following significant outcomes from the Kruskal–Wallis test, are presented using the Compact Letter Display (CLD) method. Distinct letters (e.g., A, B, and C) indicate statistically significant differences between groups, as determined by Dunn’s test with Holm–Bonferroni correction for multiple comparisons (adjusted *p* < 0.05). Identical or overlapping letters denote no significant differences between the respective groups. In this notation, groups labeled with ‘A’ exhibit significantly higher median values than those with ‘B’ (A > B), ‘A’ higher than ‘C’ (A > C), and ‘B’ higher than ‘C’ (B > C), reflecting the descending order of medians.

**Table 2 cimb-47-00868-t002:** Pairwise adjusted *p*-values for post hoc comparisons of significant parameters across PTSD groups.

Characteristic	Past PTSD (≤5 y) vs. (>5 y)	Past PTSD (≤5 y) vs. Control	Past PTSD (>5 y) vs. Control
Demographics and lifestyle			
Marital status	0.285	<0.001	<0.001
Sports performance and training	0.590	0.018	0.016
Biomarkers			
IGF-1	<0.001	<0.001	<0.001
CASP-9	<0.001	<0.001	<0.001
nNOS	<0.001	<0.001	<0.001
IL-10	<0.001	<0.001	<0.001
Coping Strategies (Brief-COPE)			
Problem-focused coping	0.180	<0.001	<0.001
Avoidant coping	0.004	<0.001	<0.001
Emotion-focused coping	<0.001	<0.001	<0.001
Overall score	0.001	<0.001	<0.001

Notes: *p*-Values are adjusted for multiple comparisons using the Holm–Bonferroni correction for continuous variables (following Kruskal–Wallis tests) and the False Discovery Rate (FDR) method for categorical variables (following Chi-square or Fisher’s exact tests). Values < 0.001 indicate highly significant differences; only parameters with significant main test results are included.

## Data Availability

The original contributions presented in this study are included in the article. Further inquiries can be directed to the corresponding author.

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
