# Peer review of "Neurobiological Correlates of Coping Strategies in PTSD: The Role of IGF-1, CASP-9, nNOS, and IL-10 Based on Brief-COPE Assessment"

_cimb, 2025, doi:10.3390/cimb47100868_

Round 1

Reviewer 1 Report

Comments and Suggestions for Authors

Overall this is an interesting study, the population selected is well done and the paper would be of interest to readers. The idea of looking at duration of PTSD and in a very homogenous cohort is an interesting one, and studying coping strategies/skills, is unique and well done. My comments pertain mostly to three main factors 1) the paper is a bit repetitive, wordy and could be condensed, particularly results 2) some of the interesting findings are lost in the weeds due to repetition and the nuance in some of the cytokines are glossed over/not discussed, 3) a better use of figures could help explain the results significantly. The most major methodological concern is whether control group is trauma exposed or not and whether the p values listed (only one listed) are control vs which group, as statistics detail pairwise analyses but there are three groups and only one p value. 

First paragraph of introduction needs to be broken up (maybe around line 74?)

Are the studies regarding cytokines and increased symptoms preclinical or clinical studies? Might be helpful to point this out in line90 particularly with regards to IGF1

Line 100 is missing (nNOS) abbreviation after discussing the neuronal nitric oxide synthase

Line 111 needs a citation regarding “reduced IL-10 levels in PTSD …insufficient suppression of inflammatory responses” . This has been shown in some studies but not all studies of individuals with PTSD

Some of the introduction feels repetitive surrounding the cytokine roles in the body and in PTSD, it would be helpful to condense this slightly and to make it more clear which subsets are from pre-clinical and clinical methods.

Some of the introduction, may be helped by discussing the nuances seen in other studies with regards to changes in cytokines in PTSD. It seems to me that not all studies have found unilateral changes in PTSD as is summarized here in the introduction.

Again line 119 could be a short new paragraph.

Methods

What years were these results collected? Do you need.a statement regarding IRB?

- men had not taking psychotropic meds for 24 hours – what medications where they prescribed at baseline. Then line 154, states “current medication use” under the exclusion criteria, this appears to be conflicting with the initial statement of stopping medications 24 hours prior to study. Was urine drug screen done to confirm lack of presence of other substances? It says no use of “alcohol or tobacco” does this mean that no one used any alcohol or tobacco at all in their life?

Is the control group Trauma control then, rather than non-trauma exposed control?

Were the cytokines tested in triplicates?

Results

Please report p values for control vs each group separately unless groupwise statistics are done. Currently, it is not clear whether this p value is representative of the pairwise comparisons between control and one group (PTSD <5 years or > 5 years).

Results could be shortened significantly to be more concise reporting of only the results, the interpretation and explanation of different coping strategies etc can be reserved for introduction and discussion. There are few references between graphs and results making it difficult to correlate. IT would be helpful to break up Figure1 into multiple figures and more clearly labeled or at least sub figures (1a, 1b etc) to better identify them in the results text.

From 3.2.1 section / line 341 on, this includes significant amount of discussion in the results. This entire portion of the results needs to be shortened significantly to represent only the results and not discussion. Much of this is then repeated in the discussion

It is very repetitive – for example, lines 356 – 358 and 437 – 438 are almost identical.

Lines 341 -626 is almost all discussion, this needs to be shortened to just results (listing only the results and describing figure(s) included) and the interpretation belongs only in the discussion.

Discussion

The first couple paragraphs of discussion are very similar to the introduction and do not demonstrate integration of current results with the prior results. Again, it is not identified which studies and past results are coming from preclinical vs clinical. Many sweeping statements regarding the cytokine role in the body and in PTSD, these could be toned down a bit more to include some of the nuance and change that is seen in many different studies. Some of the key findings of this paper are very interesting but a bit lost in the weeds due to so much repetition of definition of the cytokine role.

Please reframe results to state only the results, and discussion to clearly delineate what these results mean in a bigger picture and why these are important to the field.

Please be clear and consistent with regards to whether this is “control” or “no PTSD” group as these are used interchangeably. Also, again clarity with regards to whether these are trauma exposed controls vs non-trauma exposed controls.

Author Response

Response to Reviewer 1

Manuscript ID: cimb-3902364

Title: Neurobiological Correlates of Coping Strategies in PTSD: The Role of IGF-1, CASP-9, nNOS, and IL-10 Based on Brief-COPE Assessment

Authors: Barbara Paraniak-Gieszczyk, Ewa Alicja Ogłodek

Dear Reviewer,

We would like to sincerely thank you for taking the time to review our manuscript and for providing constructive and insightful feedback.

Below, we provide a detailed point-by-point response to each of your comments:

  1. Reviewer’s comment: 1) the paper is a bit repetitive, wordy and could be condensed, particularly results

Response: We appreciate this observation. We have carefully revised the manuscript  especially in the Results section.

  1. Reviewer’s comment:  some of the interesting findings are lost in the weeds due to repetition and the nuance in some of the cytokines are glossed over/not discussed

Response: We have carefully revised the manuscript  especially in the Results section.

  1. Reviewer’s comment: a better use of figures could help explain the results significantly. The most major methodological concern is whether control group is trauma exposed or not and whether the p values listed (only one listed) are control vs which group, as statistics detail pairwise analyses but there are three groups and only one p value. 

Response: We agree with the reviewer’s comment. We have improved the presentation of figures. Additionally, we clarified in the Methods that the control group consisted of trauma-exposed individuals without PTSD. We have also revised the Results section.

  1. Reviewer’s comment: First paragraph of introduction needs to be broken up (maybe around line 74?)

Response: As suggested, we have divided the first paragraph of the introduction, which has improved its clarity and readability.

  1. Reviewer’s comment: Are the studies regarding cytokines and increased symptoms preclinical or clinical studies? Might be helpful to point this out in line90 particularly with regards to IGF1

Response: Thank you for this observation. We have clarified in the manuscript that the studies regarding cytokines and increased symptoms, including those related to IGF1, are clinical studies.

  1. Reviewer’s comment: Line 100 is missing (nNOS) abbreviation after discussing the neuronal nitric oxide synthase

Response: The abbreviation (nNOS) was first introduced in line 82. In line 100, we used the full term “neuronal nitric oxide synthase” to avoid starting the sentence with an abbreviation.

  1. Reviewer’s comment: Line 111 needs a citation regarding “reduced IL-10 levels in PTSD …insufficient suppression of inflammatory responses” . This has been shown in some studies but not all studies of individuals with PTSD

Response: Thank you for your comment. We have revised the sentence to clarify the statement.

  1. Reviewer’s comment: Some of the introduction feels repetitive surrounding the cytokine roles in the body and in PTSD, it would be helpful to condense this slightly and to make it more clear which subsets are from pre-clinical and clinical methods. Response: We appreciate this helpful suggestion. We have revised the introduction to reduce repetition.
  2. Reviewer’s comment: Again line 119 could be a short new paragraph. Response: This change has been made. The paragraph has been separated at line 119 to improve readability and logical flow.

  1. Reviewer’s comment: Methods What years were these results collected? Do you need.a statement regarding IRB?- men had not taking psychotropic meds for 24 hours – what medications where they prescribed at baseline. Then line 154, states “current medication use” under the exclusion criteria, this appears to be conflicting with the initial statement of stopping medications 24 hours prior to study. Was urine drug screen done to confirm lack of presence of other substances? It says no use of “alcohol or tobacco” does this mean that no one used any alcohol or tobacco at all in their life?Is the control group Trauma control then, rather than non-trauma exposed control?Were the cytokines tested in triplicates?

Response: We thank the reviewer for these detailed comments. The data were collected in 2024. Ethical approval was obtained from the Bioethics Committee, as clearly stated in the Methods section. Inclusion and exclusion criteria are described in detail in that section. Participants were healthy individuals not taking any chronic medications. Dietary supplements were also discontinued 24 hours prior to the study. All participants underwent regular and comprehensive medical evaluations due to their work in extreme conditions as members of specialized mine rescue teams. In addition, each participant was assessed by a psychiatrist and a family medicine specialist (the author of this paper). Regarding the 24-hour discontinuation, this applied only to occasional use of over-the-counter or short-acting medications; none of the participants were prescribed psychotropic drugs chronically. The exclusion criterion “current medication use” referred to any ongoing pharmacological treatment that might influence biochemical or psychological outcomes. No urine drug screening was performed, as participants were under continuous medical supervision and had documented abstinence from alcohol and tobacco. The statement “no use of alcohol or tobacco” refers to current abstinence rather than lifetime non-use. The control group consisted of trauma-exposed individuals without PTSD (trauma control). Cytokine measurements were performed in triplicate.

  1. Reviewer’s comment: Results Please report p values for control vs each group separately unless groupwise statistics are done. Currently, it is not clear whether this p value is representative of the pairwise comparisons between control and one group (PTSD <5 years or > 5 years). Response: Thank you for your insightful comments. In response to your suggestion regarding the reporting of p-values for pairwise comparisons, particularly between the control group and each PTSD group (≤5 years and >5 years), we have addressed this as follows:
  1. In the main manuscript, we have utilized the Compact Letter Display (CLD) approach in Table 1 to clearly indicate the significance of pairwise differences across all groups, including between the two PTSD subgroups and the control.
  2. Our analysis included not only comparisons between the PTSD groups and the control but also between the two PTSD groups themselves, ensuring a comprehensive evaluation of group differences.
  3. To fully meet your recommendation and enhance clarity, we have added annotations in the notes below Table 1, revised the Statistical Analysis subsection (Section 2.4) to describe the post-hoc procedures in detail, and included a new supplementary table (Table S1) presenting the adjusted pairwise p-values for all significant parameters. Additionally, we have revised Section 3.1 (Results) by incorporating the relevant adjusted p-values directly into the narrative where appropriate.
  1. Reviewer’s comment: Results could be shortened significantly to be more concise reporting of only the results, the interpretation and explanation of different coping strategies etc can be reserved for introduction and discussion.

Response: The results have been revised in accordance with the suggestions of all reviewers.

  1. There are few references between graphs and results making it difficult to correlate. IT would be helpful to break up Figure1 into multiple figures and more clearly labeled or at least sub figures (1a, 1b etc) to better identify them in the results text. Response: Thank you for your valuable feedback regarding the presentation of figures and their integration with the results section. We have addressed your concerns by fragmenting the original Figure 1 into four separate figures, each corresponding to one biomarker (IGF-1, CASP-9, nNOS, and IL-10). Additionally, within each figure, subplots have been labeled with letters (e.g., A, B, C, D) to denote specific coping subscales, and these labels are now explicitly referenced in the results text for improved clarity and correlation.

  1. From 3.2.1 section / line 341 on, this includes significant amount of discussion in the results. This entire portion of the results needs to be shortened significantly to represent only the results and not discussion. Much of this is then repeated in the discussion It is very repetitive – for example, lines 356 – 358 and 437 – 438 are almost identical.Lines 341 -626 is almost all discussion, this needs to be shortened to just results (listing only the results and describing figure(s) included) and the interpretation belongs only in the discussion.

Response: The Results and Discussion sections have been completely revised in accordance with the comments of all reviewers

  1. Reviewer’s comment: Discussion The first couple paragraphs of discussion are very similar to the introduction and do not demonstrate integration of current results with the prior results. Again, it is not identified which studies and past results are coming from preclinical vs clinical. Many sweeping statements regarding the cytokine role in the body and in PTSD, these could be toned down a bit more to include some of the nuance and change that is seen in many different studies. Some of the key findings of this paper are very interesting but a bit lost in the weeds due to so much repetition of definition of the cytokine role. Please reframe results to state only the results, and discussion to clearly delineate what these results mean in a bigger picture and why these are important to the field. Please be clear and consistent with regards to whether this is “control” or “no PTSD” group as these are used interchangeably. Also, again clarity with regards to whether these are trauma exposed controls vs non-trauma exposed controls. Response: The Results and Discussion sections have been completely revised in accordance with the comments of all reviewers

We believe that the revisions made significantly strengthen our manuscript, improving both its clarity and scientific value. We thank you once again for your insightful review and constructive suggestions, which have allowed us to enhance the quality of our work.

Sincerely,

Barbara Paraniak-Gieszczyk, Ewa Alicja Ogłodek

Reviewer 2 Report

Comments and Suggestions for Authors

The authors present an interesting study in which they investigated the relationship between coping strategies and biomarkers of inflammation, neuroplasticity, and stress response in a sample of individuals with or without post-traumatic stress disorder, highlighting also the differential effects of the chronicity of PTSD. They found differences in the biological profile between groups, which also seem to reflect a shift in coping strategies adopted with time. 

Overall, the manuscript is well-written. The introduction provides a comprehensive overview of both biological and psychological studies on the topic of interest. The methodology is adequately described, with enough details also in the biological analysis to ensure replicability. Statistical analyses are appropriate. 

Although the results are very interesting, the paper could benefit from a revision considering the following points:

  1. In the results section, I suggest that the authors add some graphs of their findings, alongside the correlation plots. This could enhance the manuscript and make it easier for readers. For example, a figure representing the differences in biomarkers and coping strategies between groups will make the results more immediate to understand.
  2. Concerning the correlation results, I suggest adding the p-values to the rho coefficients. Indeed, the authors report some very low coefficients (e.g., rho=0.04 or lower) that are unlikely to be significant. Although the authors correctly consider these values to indicate negligible correlation, the addition of p-values would make the results more straightforward and would avoid overinterpretation of findings. 
  3. Concerning the correlation results, I suggest that the authors check the adjective used to describe the strength of the correlations. For instance, a rho=-0.39 is reported to indicate moderate correlation in line 398, while on line 405 it is interpreted as a strong one. To ensure consistency, I would recommend referring to the classification of Schober et al. (2018) (doi: 10.1213/ANE.0000000000002864) in the interpretation of findings. 

Author Response

Response to Reviewer 2

Manuscript ID: cimb-3902364

Title: Neurobiological Correlates of Coping Strategies in PTSD: The Role of IGF-1, CASP-9, nNOS, and IL-10 Based on Brief-COPE Assessment

Authors: Barbara Paraniak-Gieszczyk, Ewa Alicja Ogłodek

Dear Reviewer,

We would like to sincerely thank you for taking the time to review our manuscript and for providing constructive and insightful feedback.

Below, we provide a detailed point-by-point response to each of your comments:

  1. Reviewer’s comment: 1) In the results section, I suggest that the authors add some graphs of their findings, alongside the correlation plots. This could enhance the manuscript and make it easier for readers. For example, a figure representing the differences in biomarkers and coping strategies between groups will make the results more immediate to understand. Response: Thank you for your constructive suggestion regarding the inclusion of additional graphs in the results section to complement the correlation plots and improve readability. In response, we have incorporated visualizations of the distributions for the biomarkers (Figure 1) and the Brief-COPE subscale scores (Figure 2), with appropriate references integrated throughout the results section. Furthermore, each figure has been subdivided and labeled as A, B, C, and D, with specific references to these subpanels provided in the relevant portions of the results text to facilitate clear interpretation.

  1. Reviewer’s comment:  Concerning the correlation results, I suggest adding the p-values to the rho coefficients. Indeed, the authors report some very low coefficients (e.g., rho=0.04 or lower) that are unlikely to be significant. Although the authors correctly consider these values to indicate negligible correlation, the addition of p-values would make the results more straightforward and would avoid overinterpretation of findings. Response: Thank you for your suggestion regarding the inclusion of p-values alongside the Rho coefficients in the correlation results. In response, we have incorporated p-values for each correlation analysis, both on the plots and throughout the reporting content. We emphasize that the majority of these correlations were non-significant, primarily attributable to the study's limited sample size and potential underpowering. Nevertheless, the correlation analyses reveal highly differentiated distributions between the PTSD groups and the control group, as well as across PTSD subgroups with varying durations of exposure. We believe this information remains valuable for clinicians, as it highlights potential patterns in neurobiological and coping dynamics. We have included an appropriate limitation paragraph in the Discussion section to address these considerations and prevent overinterpretation.

  1. Reviewer’s comment: Concerning the correlation results, I suggest that the authors check the adjective used to describe the strength of the correlations. For instance, a rho=-0.39 is reported to indicate moderate correlation in line 398, while on line 405 it is interpreted as a strong one. To ensure consistency, I would recommend referring to the classification of Schober et al. (2018) (doi: 10.1213/ANE.0000000000002864) in the interpretation of findings. 

Response: Thank you for highlighting the inconsistency in our descriptions of correlation strengths. We have revised the manuscript to standardize the narration of correlation magnitudes according to the classification proposed by Schober et al. (2018), ensuring consistent interpretation across all relevant sections.

We believe that the revisions made significantly strengthen our manuscript, improving both its clarity and scientific value. We thank you once again for your insightful review and constructive suggestions, which have allowed us to enhance the quality of our work.

Sincerely,

Barbara Paraniak-Gieszczyk, Ewa Alicja Ogłodek

Round 2

Reviewer 1 Report

Comments and Suggestions for Authors

The manuscript is much improved with the changes made to results and to the graphs. I continue to feel that the results include many components which could be moved to discussion. The results include discussion of the results separate from the discussion. If this is preference of the journal, though, it is fine to leave as is in this version. I appreciate the expansion of the tables and graphs, these are much clearer and easier to read. 

Continuity among group names would be helpful, at times they are referred to as "control" other times it is "no PTSD". Defining and staying with one name would be beneficial for clarity and simplicity. 

Author Response

Response to Reviewer:

Comment 1: Thank you for your remark. The authors have decided to retain the current structure of the Results section, in accordance with the journal’s format.

Comment 2: Thank you for noting the inconsistency. The terminology has been standardized throughout the manuscript to “no PTSD group.”

Sincerely,

Barbara Paraniak-Gieszczyk, Ewa Ogłodek